# Risk Factors for Lung Function Decline in Pediatric Asthma under Treatment: A Retrospective, Multicenter, Observational Study

**DOI:** 10.3390/children9101516

**Published:** 2022-10-04

**Authors:** Shingo Yamada, Takao Fujisawa, Mizuho Nagao, Hiroshi Matsuzaki, Chikako Motomura, Hiroshi Odajima, Toshinori Nakamura, Takanori Imai, Ken-ichi Nagakura, Noriyuki Yanagida, Masatoshi Mitomori, Motohiro Ebisawa, Shigenori Kabashima, Yukihiro Ohya, Chizu Habukawa, Minako Tomiita, Masahiro Hirayama

**Affiliations:** 1Allergy Center, National Hospital Organization Mie National Hospital, Tsu 514-0125, Japan; 2Allergy Center, National Hospital Organization Mie National Hospital, 357 Ozato-kubota, Tsu 514-0125, Japan; 3Department of Pediatrics, National Hospital Organization Fukuoka National Hospital, Fukuoka 811-1394, Japan; 4Department of Pediatrics, Showa University School of Medicine, Tokyo 142-8666, Japan; 5Department of Pediatrics, National Hospital Organization Sagamihara National Hospital, Sagamihara 252-0392, Japan; 6Allergy Center, National Center for Child Health and Development, Tokyo 157-8535, Japan; 7Department of Pediatric Allergy, National Hospital Organization Minami Wakayama Medical Center, Tanabe 656-8558, Japan; 8Center of Pediatric Allergy and Rheumatology, National Hospital Organization Shimoshizu National Hospital, Yotsukaido 284-0003, Japan; 9Department of Pediatrics, Mie University Graduate School of Medicine, Tsu 514-8507, Japan

**Keywords:** asthma, childhood and adolescence, asthma control, inhaled corticosteroids, lung function, risk factors

## Abstract

Background: Childhood asthma is a major risk for low lung function in later adulthood, but what factors in asthma are associated with the poor lung function during childhood is not known. Objective: To identify clinical factors in children with asthma associated with low or declining lung function during the treatment. Methods: We enrolled children with asthma who had been treated throughout three age periods, i.e., 6–9, 10–12, and 13–15 years old, at seven specialized hospitals in Japan. Clinical information and lung function measurements were retrieved from the electronic chart systems. To characterize the lung function trajectories during each age period, we evaluated the forced expiratory volume 1 (FEV1) with % predicted values and individual changes by the slope (S) from linear regression. We defined four trajectory patterns: normal (Group N) and low (Group L), showing %FEV1 ≥80% or <80% throughout all three periods; upward (Group U) and downward (Group D), showing S ≥ 0 or S < 0%. Logistic regression analysis was performed to compare factors associated with the unfavorable (D/L) versus favorable (N/U) groups. Results: Among 273 eligible patients, 197 (72%) were classified into Group N (*n* = 150)/U (*n* = 47), while 76 (28%) were in Group D (*n* = 66)/L (*n* = 10). A history of poor asthma control, long-acting beta2 agonist use, and a lower height Z-score during 13–15 years were associated with an unfavorable outcome (Group D/L). Conversely, inhaled corticosteroid (ICS) use during 10–12 years and high-dose ICS use during 13–15 years were associated with a favorable outcome (Group N/U). Conclusion: We identified several factors that are associated with unfavorable lung function changes in pediatric asthma. Attention should be paid to the possible relationship between yearly changes in lung function and poor asthma control, use of ICS (and its dose) and use of LABA.

## 1. Introduction

Lung function develops during childhood and reaches its peak in early adulthood. That peak has been reported to have a potential impact on the subsequent development or progression of various diseases, not only in the respiratory system, but also in the cardiovascular and other systems, even influencing life expectancy [1,2,3]. Thus, the lung function status in childhood is critically important for long-term health. Poor lung development during childhood may be due to perinatal factors, such as noxious exposures in utero [4], preterm birth [5,6,7,8], and postnatal injury to the lungs, due to early respiratory infections [9,10], asthma [11], etc.

Population-based birth cohort studies investigated a broad range of possible risk factors for lung function decline and found that asthma and its related characteristics, such as wheezing, bronchial hyperresponsiveness, and aero-allergen sensitization, are major contributors [12]. When the focus was on asthma, about one-fourth of children with mild or moderate asthma who participated in a 4-year clinical trial of anti-inflammatory treatment (CAMP study) [13] were reported to have a significant (≥1% per year) decline in forced expiratory volume in 1 s (FEV1), regardless of the treatment assigned, i.e., inhaled corticosteroid (ICS), nedocromil, or placebo [14]. Although the continuous use of ICS was associated with a smaller annual decline of FEV1 in children [15] and adults [16] with asthma, we still have a poor understanding of the roles of such detailed factors as the ICS dosage, use of other medications, the degree of asthma control, etc.

In order to prevent lung function decline in pediatric patients with asthma and improve their long-term outcome, it is important to identify the modifiable factors involved in the decline. Therefore, we explored the risk factors for lung function decline in pediatric patients with asthma who had been treated at specialized hospitals, where the best possible care was supposedly provided, yet some patients, nevertheless, had low lung function. We hope that our findings will help clinicians decide the optimal treatments for their pediatric asthma patients.

## 2. Methods

### 2.1. The Study Population

This was a retrospective multicenter observational study that enrolled patients with pediatric asthma who were treated between 2006 to 2019 at 7 specialized hospitals in Japan (National Hospital Organization Mie National Hospital; National Hospital Organization Fukuoka National Hospital; National Hospital Organization Sagamihara National Hospital; National Hospital Organization Shimoshizu National Hospital; National Hospital Organization Minami Wakayama Medical Center; National Center for Child Health and Development; and Department of Pediatrics Showa University School of Medicine). Eligible patients were those: (1) who had been regularly treated for asthma for at least 5 years during the period from 6 to 15 years of age; (2) whose spirometry was performed at least once during each of 3 age intervals: pre-adolescence (6 to 9 years of age), early adolescence (10 to 12 years of age), and adolescence (13 to 15 years of age).

### 2.2. Data Collection

The following clinical data were retrieved from electronic charts: birth weight, gestational age, history of comorbid allergic diseases, such as food allergy (FA), atopic dermatitis (AD), perennial allergic rhinitis (PAR) and seasonal allergic rhinitis (SAR), other comorbidities, family history of asthma, parental smoking, pet ownership, results of spirometry, height and weight at each measurement, previous administration of inhaled corticosteroids (ICS) and leukotriene receptor antagonist (LTRA) before 6 years of age, use of ICS and its dose, use of LTRA and long-acting beta-adrenoreceptor antagonist (LABA) at each visit, fractional exhaled nitric oxide (FeNO), asthma control levels by childhood asthma control test (C-ACT) [17] during the age period of 6–11 years and asthma control test (ACT) [18] during 12–15 years, total-IgE (IU/mL), house dust mite-specific IgE (HDM-sIgE) (U_A_/mL), and Japanese cedar-pollen-specific IgE (JCP-sIgE) (U_A_/mL) (Immunocap^®^; Thermo Fisher Diagnostics). Non-adherence recorded by the attending pediatricians was also retrieved. Spirometry results were evaluated as a 1-year average of percent predicted forced expiratory volume in one second (%FEV1), in which reference values were calculated by the equations for Japanese children [19]. Since the study institutions were secondary/tertiary centers, some patients were regularly treated at primary institutions and periodically referred from the institutions, and intervals of spirometry, i.e., intervals of patient visits, varied. Numbers of spirometry measurements were recorded and summarized in Appendix A. Loss of asthma control was defined as C-ACT or ACT ≤ 19.

### 2.3. Lung Function

The change in %FEV1 was estimated for each subject by linear regression of each measured data. The slope (S) was calculated from the fitted line, with an equation of y = α + βx, in which x is the number of years from the initial measurement. We classified the changes in %FEV1 into 4 trajectories. The subjects with %FEV1 ≥ 80, or above lower limit of normal (LLN) [19], throughout the observation period, were classified as Group N (normal trajectory), and those with %FEV1 < 80, or below LLN, throughout the observation, were classified as Group L (low trajectory). Then, among the patients who had %FEV1 < 80 in any year of follow-up, those with S ≥ 0 were classified as Group U (upward trajectory), and those with S < 0 were classified as Group D (downward trajectory). In terms of lung function growth, FEV1 in Group N was considered to have grown or increased, as normal values increased with physical growth. FEV1 in Group U was considered to have moved from relative decline to normal growth. Thus, favorable trajectories, i.e., Groups N and U, were then combined as Group N/U. FEV1 in Group L was considered to have not grown, as the normal counterpart increased and FEV1 in Group D was considered to have further declined. Then, unfavorable trajectories, i.e., Groups L and D, were combined as Group L/D for analysis of risk (Figure 1).

### 2.4. Statistical Analyses

Categorical variables were compared using the chi-square test. Two groups of continuous variables were compared using the Mann–Whitney test, and 4 groups of continuous variables were compared with the Kruskal–Wallis test, followed by the Dunn’s multiple comparison test, to compare each group. Height and body mass index (BMI) were converted into Z-scores and compared with the Japanese population using a one-sample *t*-test [20,21]. Risk factors for unfavorable lung function trajectories were analyzed by multivariate logistic regression analysis. Statistical tests were performed using JMP version 16.1 (SAS Institute, Cary, NC, USA) and GraphPad Prism version 9.3.1 (GraphPad Software, San Diego, CA, USA).

### 2.5. Ethics

The study was approved by the Ethics Committee of National Hospital Organization Mie National Hospital (approval number: 31–81), and permission to obtain data from the electronic medical systems and merge the data from the 7 contributing hospitals was granted.

## 3. Results

### 3.1. Classification of %FEV1 Trajectories

A total of 628 patients were enrolled, and 354 patients were excluded, due to lack of spirometry data. A biologic, namely omalizumab, was administered to only two patients. We excluded them from the analyses because the effect of biologics on lung function trajectories is potentially strong. Then, data from 273 children/adolescents with asthma were analyzed. There were no differences in the clinical backgrounds between patients who were included and excluded for analysis (Appendix A).

In accordance with our above definitions, 150 patients (55%) were classified in Group N, with 47 (17%) in Group U, 66 (24%) in Group D, and 10 (4%) in Group L. Figure 1 shows the classification and changes in mean %FEV_1_ from 6 to 15 years of age in the four trajectory groups.

The mean (SD) annual changes in %FEV1 were −0.78 (−1.0), 3.7 (2.8), −1.9 (−1.7), and −0.02 (−0.08) %/year in Groups N, U, D, and L, respectively. The decline in %FEV1 in Group D was significantly larger than in all the other groups (Figure 2). By definition, the %FEV1 was normal (≥80%) throughout the observation period in Group N, but an annual decline of around −1% was observed.

### 3.2. Clinical Characteristics of the Subjects

Table 1 summarizes the clinical background of the subjects. There were no differences in gender ratio (male predominance), median gestational age, or birth weight among the four trajectory groups. Each of the groups included small numbers of subjects with preterm birth and/or low birth weight. The prevalences of comorbid allergic diseases, parental asthma, parental smoking, and pet ownership were similar among the groups, but FA was higher in Groups N and D (Table 1). The maximal values of the blood eosinophil count, total IgE, HDM−sIgE, and JCP−sIgE during the observation period were similar among the groups, with no statistical differences. The prevalence of asthma treatment before 6 years of age was also similar among the groups (Table 1).

Table 2 shows the anthropometric data of the subjects. The Z-scores for height and BMI were negative in Groups N, D, and L, but positive in Group U. The Z-scores for height were significantly low in the 10–12 and 13–15 age periods in Group D and the 13–15 age period in Group N. Similarly, the Z-scores for BMI were significantly low in Groups N and D for all age categories (Table 2).

### 3.3. Factors Associated with Unfavorable Lung Function Outcome

The subjects of this study showed no differences in the risk factors previously reported for low lung function, such as low birth-weight [22,23], comorbid allergic rhinitis [24], obesity [23], passive smoking [25,26], and allergen sensitization [12]. Accordingly, we thereafter focused on asthma control and treatment. We compared the level of asthma control, FeNO, and treatment in the favorable Group N/U and unfavorable Group D/L during the three age periods, i.e., 6–9, 10–12, and 13–15 years. The number of patients who experienced loss of asthma control at any time during each period, represented as C-ACT or ACT ≤ 19, was significantly higher in Group D/L than Group N/U in the 10–12 and 13–15 age periods (Table 3). The mean FeNO value in each age period showed no differences between Groups D/L and N/U. Most of the patients were treated with ICS, and a small population received a high dose of ≥400 µg/day; the proportions did not differ between Groups D/L and N/U. The proportion of patients treated with a long-acting beta2 agonist (LABA) was significantly higher in Group D/L in the 10–12 and 13–15 age periods than in Group N/U. There was no difference in use of LTRA (Table 3). Of note, non-adherence rate was adequately small at 9 in 273.

A logistic model for predicting an unfavorable lung function outcome was constructed using variables that were shown to be significantly different by univariate analyses, as well as those considered clinically important: gender, Z-scores for height and BMI, ACT/c-ACT ≤ 19, LABA use, ICS use, and high-dose ICS use (fluticasone-equivalent dose of ICS ≥ 400 µg/day) during each age period. Logistic analysis identified a lower Z-score for height, ACT ≤ 19, use of LABA, use of high-dose ICS in the 13–15 age period, and use of ICS in the 10–12 age period as statistically significant factors for predicting the outcome (Table 4). The odds ratio (OR) for the height Z-score was 0.65, indicating that a short stature was associated with unfavorable trajectories. The ORs for ICS use and high-dose ICS use were low, i.e., 0.27 and 0.17, while those for ACT ≤ 19 and LABA use were significantly high, indicating that the former two factors may be associated with a favorable outcome, and the latter two may be associated with an unfavorable outcome.

## 4. Discussion

In this study, we analyzed the long-term changes in FEV1 in pediatric patients with asthma under treatment at specialized hospitals. We found that 28% of them had low or declining lung function trajectories, and the unfavorable trajectories were associated with temporary loss of asthma control, LABA use, and a low Z-score for height in adolescence. On the other hand, ICS use in early adolescence and high-dose ICS use in adolescence were associated with opposite outcomes. Although the long-term implications of the current findings remain to be clarified, the findings may provide clues regarding how to intervene to reverse declining lung function in children and adolescents.

Characterization of lung function trajectories over a lifetime has received a great deal of attention recently. In the normal population, the lung function trajectory shows three phases: the growth, plateau, and decline phases [27]. Impaired lung function trajectories result from poor growth in the first phase, a low peak in the second phase, and/or rapid decline beyond the normal aging process in the third phase. In this study, we focused on the first phase in pediatric asthma because it is a major risk factor for poor lung function development, leading to low peak lung function in early adulthood [12].

Several birth cohort studies identified a neonatal origin of low lung function in later life. The Perth Infant Asthma Follow-up (PIAF) study [28] in Australia showed that a low maximum flow rate at functional residual capacity (V’maxFRC) at 1 month was significantly associated with persistent wheeze at 11 years [29]. Additionally, infants with a low V˙maxFRC below the median had significantly lower forced expiratory flow at 25–75% of vital capacity (FEF25–75%) at 6, 11, 18, and 24 years, and lower FEV1/FVC at 11 and 24 years, compared with infants with higher V’maxFRC [30]. Preterm infants often have a number of respiratory problems during the neonatal period and are demonstrated to have long-term sequelae of impaired lung function [31,32]. On the other hand, infants in the PIAF cohort were full-term and had no perinatal problems [28]. Likewise, the Tucson Children’s Respiratory Study (TCRS) enrolled 1246 healthy infants at birth, and 376 of them were tested for V’maxFRC. Lower quartiles of infant V’maxFRC tracked to 16- and 22-year-olds having low %FEV1, low %FEV25–75%, and low FEV1/FVC ratio [33]. Those results suggest that *in-utero* alterations in airway development may predispose lung function deficits in adolescent and adult life. For our asthma patient cohort, however, we had no information on lung function in infancy. The early identification of poor lung function in infancy might help us predict the long-term lung function outcome in individual patients.

Focusing on asthma, the CAMP study found that, in 25.7% of children, the lung function declined by more than 1% per year during 4 years of randomized treatment [14]. We found a similar rate, 28%, in this study. The predictors in a CAMP study subgroup, defined as SRP (significant reduction of postbronchodilator FEV1% at baseline), were younger age, male sex, and higher postbronchodilator FEV1%. The percentage of patients classified as SRP was similar in each of the CAMP study treatment groups, i.e., budesonide, nedocromil, and placebo, suggesting that anti-inflammatory treatment, especially with ICS, did not prevent lung function decline [14]. On the other hand, among 119 allergic asthmatic subjects aged 5 to 14 years who were followed up to ages 22 to 32 and 32 to 42 years in the Netherlands, who quit smoking, and continued to use ICS had a significantly smaller annual decline in FEV1 during adulthood [15], suggesting a favorable effect of ICS. Furthermore, in an adult asthma cohort in the European Community Respiratory Health Survey, the FEV1 decline was lower in subjects who had used ICS for a longer time [16]. In our study, the use of ICS—especially at a high dose—was negatively associated with declining or low lung function. Although it is unclear whether a “sufficient” ICS dose is effective in preventing lung function decline, this warrants further investigation.

Physical development may affect the growth of lung function. Preterm birth is a major negative factor for lung growth [8,31,32], but catch-up physical growth from birth to 5 years in preterm infants can lead to a favorable outcome, i.e., higher FEV1 at 21 years of age [34]. On the other hand, in a birth cohort study including children with a family history of allergic disease (the Melbourne Atopy Cohort study, MACS) who were not preterm babies [35], the “early-low and catch-up” and “persistently high” BMI trajectories were at higher risk of asthma at the age of 18 years, with a low FEV1/FVC ratio [36]. These seemingly contradictory reports indicate that prematurity and overweightness—the two extreme ends in child anthropometry—each have a negative impact on lung function growth. In our asthma patient cohort, the Z-scores for BMI and height were negative in all age ranges and all trajectory groups, except the upward trajectory group (Table 2), indicating that the subjects had slightly small stature, which may have been related to their low lung function. It seems clear that physical growth should also be evaluated in assessing changes in lung function, although pathophysiological significance of slightly short height needs further investigation

Long-term follow-up of the CAMP cohort identified factors in childhood that predicted severe asthma at 17–19 and 21–23 years [37]. First, it was found that every 5% decrease in the post-bronchodilator FEV1/FVC ratio in childhood increased the odds of severe asthma during both age ranges by 2.36-fold. Second, the frequency of emergency visits for asthma during childhood was significantly higher in patients with severe asthma during late adolescence and early adulthood. The latter finding corresponds, at least partially, to our current finding that poor asthma control (C-ACT or ACT ≤ 19) was associated with unfavorable lung function trajectories. The subjects in our study had been treated at specialized hospitals, where guideline-based, best-possible treatment was provided; their asthma was mostly under control, yet occasional deterioration of control occurred. Likewise, the CAMP study subjects had mild or moderate asthma that was mostly controlled. Collectively, we assume that even short-term exacerbations may warrant more intensive treatment to prevent lung function decline.

Systematic reviews indicated that the addition of LABA to ICS improved lung function in pediatric asthma [38,39]. However, in our present study, LABA use was associated with unfavorable lung function trajectory. These conflicting results can, perhaps, be explained as follows. First, the studies evaluated in the reviews tested the effect of LABA on lung function during randomized phases, not long-term changes. Second, the apparent stability of symptoms with LABA may have masked the progression of lung function decline, due to airway inflammation, which cannot be controlled by LABA. Thirdly, conversely, our results may arise simply by selection bias or reverse causality, i.e., LABA was administered to severe patients with lung function decline, due to other factors, not LABA.

Our study has a number of limitations. First, although the study examined changes in lung function over a period of 5 or more years, this retrospective cohort did not contain a group of patients whose symptoms had improved and stopped visiting hospitals. In addition, our study population may not represent the whole pediatric asthma population, since our institutions provide secondary/tertiary care and we usually refer patients back to referring primary institutions after short period of evaluation and establishment of a treatment. Factors associated with lung function trajectory might have been more clearly identified if such a patient group had been included. However, we focused on patients who needed special attention, i.e., they required ongoing treatment at secondary/tertiary institutions and were at risk of lung function deterioration. Second, the trajectory was evaluated only during childhood, and final outcomes must be assessed in adulthood. Further longitudinal observation is necessary. Third, and most importantly, the risk factors identified in this study were merely shown to be associated with longitudinal changes in lung function, and causal relationships in the sequence of time were not demonstrated. For example, LABA use was a risk for an unfavorable lung function trajectory, whereas high-dose ICS was conversely demonstrated to reduce the risk; however, it is uncertain whether lung function decline would have been prevented if high-dose ICS had been administered, instead of LABA [39]. We think that our results may, at least, lead clinicians to consider administering high-dose ICS to patients who would benefit from it. Fourth, use of a “sufficient” dose of ICS to prevent lung function decline has to be balanced with the potential negative effect of adrenal suppression [40].

In conclusion, we identified several factors that seem to be associated with unfavorable lung function changes in pediatric asthma. Attention should be paid to the possible relationship between yearly changes in lung function and poor asthma control, as well as the use of ICS and LABA. Future prospective studies should clarify whether more aggressive use of ICS, while avoiding over-reliance on LABA, can prevent not only asthma exacerbations in childhood, but also lung function decline in later life.

## Figures and Tables

**Figure 1 children-09-01516-f001:**
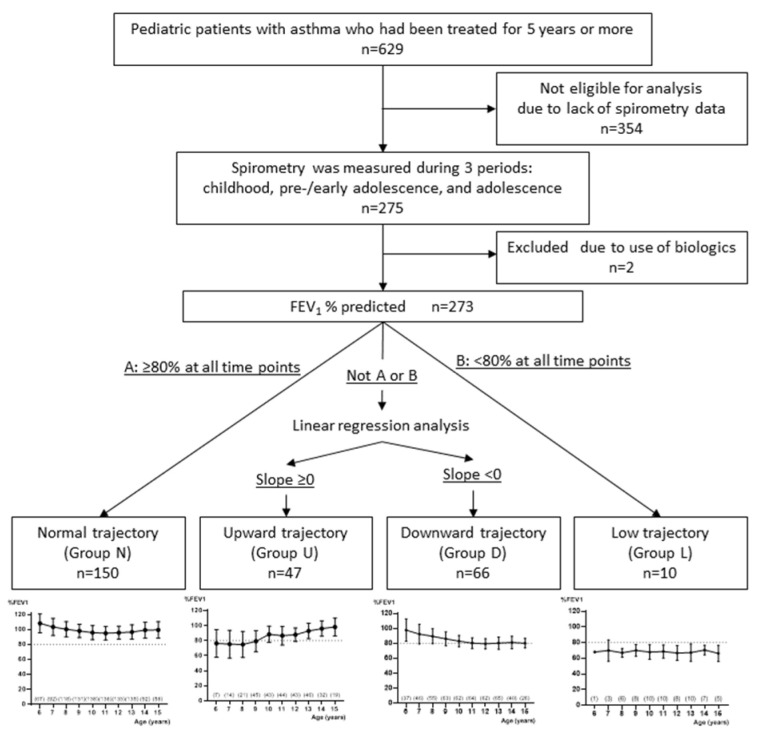
Flow of the subjects and classification of lung function trajectories in the retrospective study. Numbers in parenthesis indicate the number of patients at each age.

**Figure 2 children-09-01516-f002:**
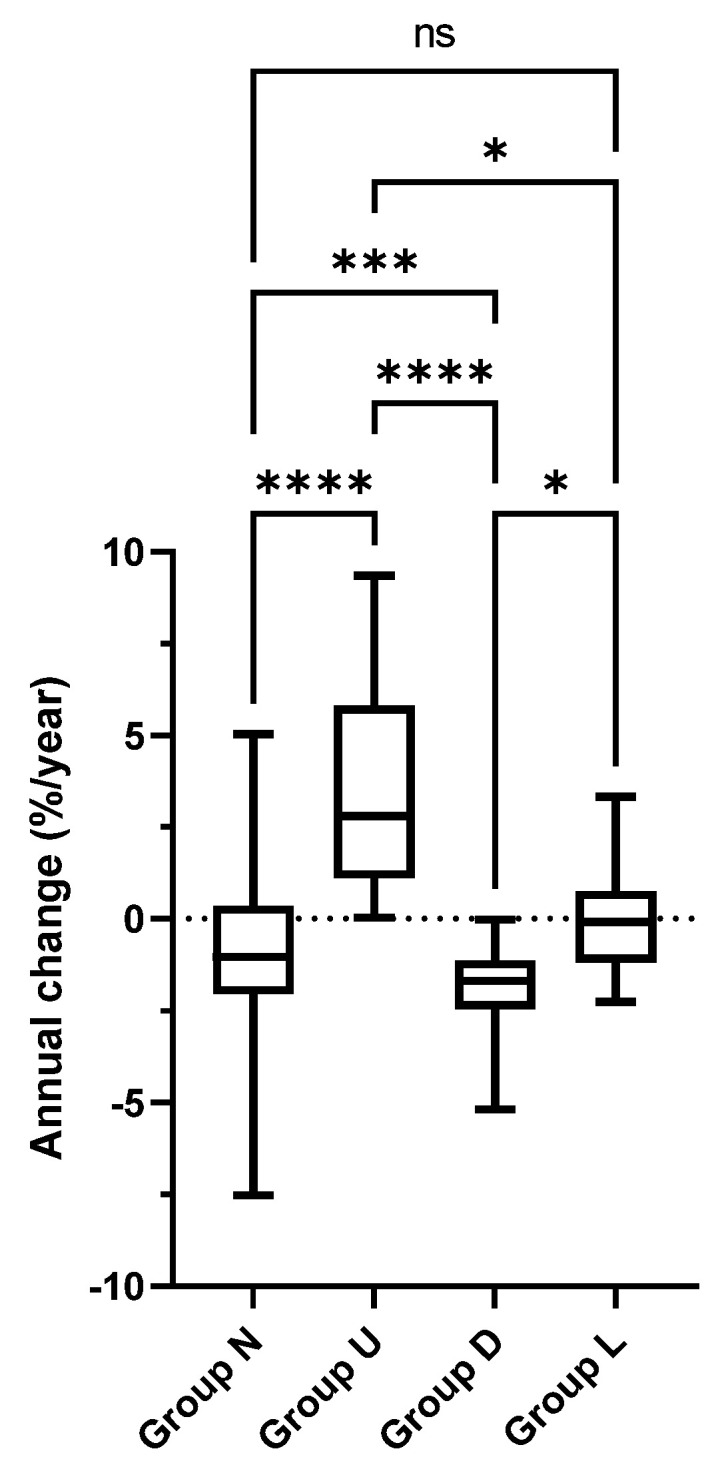
Annual changes in %FEV1 in lung function trajectory groups. *p* < 0.0001 with Kruskal–Wallis test. **** *p* < 0.0001, *** *p* < 0.001, and * *p* < 0.05 with Dunn’s multiple comparison test. n.s.: not significant.

**Table 1 children-09-01516-t001:** Clinical background of the subjects.

Background Factor	*n* ^#1^	Trajectory Type				
		Normal *n* = 150	Upward *n* = 47	Downward *n* = 66	Low *n* = 10	*p* Value
Sex (boys), *n* (%)	273	89 (59)	30 (64)	43 (65)	6 (60)	0.853
Gestational age (weeks), median (range)	210	39 (30–42)	39 (28–41)	39 (30–42)	38 (24–41)	0.698
Birth weight (g), median (range)	220	2968 (860–4494)	3036 (1158–3896)	3060 (1372–4050)	3016 (845–3966)	0.671
Comorbid allergic diseases						
Atopic dermatitis, *n* (%)	272	79 (53)	23 (49)	28 (43)	3 (30)	0.367
Perennial allergic rhinitis, *n* (%)	272	92 (61)	31 (66)	31 (48)	5 (50)	0.172
Seasonal allergic rhinitis, *n* (%)	272	42 (28)	14 (30)	17 (26)	2 (20)	0.923
Food allergy, *n* (%)	270	78 (52)	17 (37)	32 (49)	1 (10)	0.027
Other comorbidities	273	13 (9)	11 (23)	7 (11)	1 (10)	0.045
Family history and environment						
Parental asthma, *n* (%)	249	30 (22)	8 (19)	9 (15)	3 (33)	0.559
Parental smoking, *n* (%)	222	43 (34)	14 (36)	20 (40)	4 (50)	0.768
Pet ownership, *n* (%)	209	33 (27)	7 (25)	8 (16)	3 (38)	0.346
Laboratory data ^#2^						
Eosinophils (/μL), median (range)	261	490 (0–3063)	524 (0–1402)	535 (100–2520)	390 (118–1040)	0.738
Total-IgE (IU/mL), median (range)	258	954 (10–12700)	876 (39–7300)	743 (4–10600)	602 (99–8726)	0.625
HDM-sIgE (kU_A_/L), median (range)	245	100 (<0.1–800)	81.9 (<0.1–553)	99.7 (<0.1–394)	58.7 (0.5–436)	0.729
JCP-sIgE (kU_A_/L), median (range)	248	31.5 (<0.1–468)	22.9 (<0.1–553)	34.7 (<0.1–477)	28.7 (<0.1–249)	0.715
Asthma treatment before 6 years old						
ICS	252	75 (54)	29 (64)	37 (65)	8 (80)	0.179
LTRA	254	113 (80)	45 (87)	62 (76)	10 (70)	0.615

^#1^ Number; ^#2^ maximal values during the observation period.

**Table 2 children-09-01516-t002:** Anthropometric data of the subjects.

	Trajectory Type			
	Normal	Upward	Downward	Low
**Z-Score for Height**				
6–9 years; mean	−0.066	0.232	−0.221	−0.121
95% CI	−0.219 to 0.0863	−0.0617 to 0.527	−0.469 to 0.0268	−0.832 to 0.590
*n*	145	46	66	10
*p*-value	n.s.	n.s.	n.s.	n.s.
10–12 years; mean	−0.133	0.302	−0.340 #	−0.133
95% CI	−0.283 to 0.0166	−0.009 to 0.613	−0.599 to −0.080	−1.012 to 0.745
*n*	147	46	65	10
*p*-value	n.s.	n.s.	0.0112	n.s.
13–15 years; mean	−0.249	0.193	−0.388 #	−0.181
95% CI	−0.399 to −0.0981	−0.109 to 0.495	−0.655 to −0.112	−1.244 to 0.882
*n*	144	46	66	10
*p*-value	0.0014	n.s.	0.0052	n.s.
**Z-Score for BMI**				
6–9 years; mean	−0.271	0.079	−0.330	−0.692
95% CI	−0.422 to −0.119	−0.248 to 0.406	−0.562 to −0.099	−1.453 to 0.070
*n*	145	46	66	10
*p*-value	0.0006	n.s.	0.0059	n.s.
10–12 years; mean	−0.296	0.011	−0.357	−0.655
95% CI	−0.448 to −0.144	−0.304 to 0.326	−0.567 to −0.147	−1.366 to 0.056
*n*	147	46	65	10
*p*-value	0.0002	n.s.	0.0012	n.s.
13–15 years; mean	−0.205	0.242	−0.280	−0.851
95% CI	−0.383 to −0.028	−0.117 to 0.602	−0.518 to −0.0416	−1.563 to −0.139
*n*	144	46	66	10
*p*-value	0.0238	n.s.	0.0221	0.0242

*p*-value is from a one-sample t-test comparing the mean Z-scores with a hypothetical normal value of 0. # *p* < 0.05 with Kruskal–Wallis test, followed by Dunn’s multiple comparison test, upward vs. downward. n.s.: not significant.

**Table 3 children-09-01516-t003:** Comparison of asthma control level, FeNO, and treatment between favorable and unfavorable lung function trajectory groups.

Factor	Trajectory Type		*p* Value ^#1^
	Normal/Upward *n* = 197	Downward/Low *n* = 76	
**C-ACT or ACT** **≤ 19, *n* (%)**			
6–9 years	53 (28)	21 (28)	n.s.
10–12 years	32 (17)	26 (35)	0.001
13–15 years	14 (7)	12 (16)	0.030
**FeNO (ppb), median (range)**			
6–9 years	33 (1–142)	37 (2–120)	n.s.
10–12 years	48 (7–197)	53 (7–99)	n.s.
13–15 years	53 (7–165)	47 (15–179)	n.s.
**Treatment**			
**ICS use**	*n* (%)		
6–9 years	154 (92)	58 (92)	n.s.
10–12 years	161 (89)	64 (90)	n.s.
13–15 years	109 (77)	44 (80)	n.s.
**High-dose ICS use** ^#2^	*n* (%)		
6–9 years	6 (4)	4 (6)	n.s.
10–12 years	17 (9)	4 (6)	n.s.
13–15 years	17 (12)	3 (6)	n.s.
LABA use	*n* (%)		
6–9 years	61 (36)	32 (49)	n.s.
10–12 years	75 (40)	44 (61)	0.002
13–15 years	51 (31)	31 (55)	0.001
**LTRA use**	*n* (%)		
6–9 years	138 (81)	53 (79)	n.s.
10–12 years	167 (88)	64 (89)	n.s.
13–15 years	93 (50)	38 (57)	n.s.

^#1^ Chi-square test; ^#2^ fluticasone-equivalent dose of ICS ≥ 400 µg/day. n.s.: not significant.

**Table 4 children-09-01516-t004:** Factors associated with unfavorable lung function trajectories by multivariate logistic regression analysis.

Factor	OR	95% CI	*p*-Value
Height Z-score in the 13–15 age period	0.65	0.45–0.93	0.016
ACT ≤ 19 in the 13–15 age period	3.87	1.29–12.32	0.016
ICS use in the 10–12 age period	0.27	0.08–0.88	0.031
High-dose ICS use in the 13–15 age period	0.17	0.03–0.67	0.010
LABA use in the 13–15 age period	2.85	1.21–6.89	0.017

OR: odds ratio, n.s.: not significant factors are binary variables, except for height Z-score, which is continuous variable.

## Data Availability

Not applicable.

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
