# Peer review of "Risk Factors for Lung Function Decline in Pediatric Asthma under Treatment: A Retrospective, Multicenter, Observational Study"

_children, 2022, doi:10.3390/children9101516_

Round 1

Reviewer 1 Report

Dear Authors,

the article is very interesing, please provide the minor corrections.

Abstract:

The background of the study is not sufficient. Please, paraphrase and expand. The undesirable outcome is not accurate term, what authors had in their mind?

Authors did not mention the aim of the study. This is the mistake.

Methods should include number of participants: does it mean that the number of children during the different periods of observation were different?

Results: This sentence: : “A total of 273 patients were enrolled, and 197 (72%) were classified into Group N (n=150)/U 32 (n=47), while 76 (28%) were in Group D (n=66)/L (n=10)” should be mentioned in the methods.

Conclusions: The conclusions are not supported with the results.

Keywords: Since you were looking for the risk factors then maybe is should be mentioned as a keyword.

Main text:

In statistical analyses: The Dunn’s correction is not mentioned.

Results: Table 1. Please, change Gender with sex term. Gender is a psychological term not epidemiological/clinical.

Gestational age (w) what “w” means?

Table 3. Those no statistical differences between FeNO are hard to believe, could authors verify it?

References should be prepared according to journal recommendations.

Author Response

COMMENTS FROM REVIEWER #1:

The article is very interesting, please provide the minor corrections.

We thank this Reviewer for his/her nice words about our work.

 Abstract:

  1. The background of the study is not sufficient. Please, paraphrase and expand. The undesirable outcome is not an accurate term, what authors had in their mind?

We thank this Reviewer for this comment. We have now specifically described what was not known in this field as follows:

“Background: Childhood asthma is a major risk for low lung function in later adulthood, but it is not known what factors in asthma are associated with the poor lung function during childhood.”

  1. Authors did not mention the aim of the study. This is the mistake.

We thank this Reviewer for this comment. We have now added the objective of the study as follows:

“Objective: To identify clinical factors in children with asthma associated with low or declining lung function during the treatment.”

  1. Methods should include number of participants: does it mean that the number of children during the different periods of observation were different?

We followed one of the reporting guidelines, STROBE, (https://www.equator-network.org/reporting-guidelines/strobe/ ) and stated the number of participants in Results section. To address the reviewer’s concern, the relevant text has been corrected as follows:

“A total of 629 patients were screened and 354 patients with lack of sufficient spirometry data were not included. Two patients treated with biologics were excluded because of possible large effect on lung function. Clinical background of the excluded patients was similar with those included. Among 273 eligible patients, 197 (72%) were~~”

  1. Results: This sentence: : “A total of 273 patients were enrolled, and 197 (72%) were classified into Group N (n=150)/U 32 (n=47), while 76 (28%) were in Group D (n=66)/L (n=10)” should be mentioned in the methods.

We describe the definition of outcomes, changes in FEV1 over years (or trajectories), in Methods section and the resulting number of the outcomes in Results section according to STROBE.

Definition of the outcomes were as follows:

We defined 4 trajectory patterns: normal (Group N) and low (Group L), showing %FEV1 ≥80% or <80% throughout all 3 periods; and upward (Group U) and downward (Group D), showing S≥0 or S<0%.”

  1. Conclusions: The conclusions are not supported with the results.

We thank the Reviewer for this important comment. We corrected the overstatement as follows:

“We identified several factors that are associated with unfavorable lung function changes in pediatric asthma. Attention should be paid to the possible relationship between yearly changes in lung function and poor asthma control, use of ICS (and its dose) and use of LABA.”

  1. Keywords: Since you were looking for the risk factors then maybe is should be mentioned as a keyword.

According to the reviewer’s comment, we added “risk factors” in Keywords list.

Main text:

  1. In statistical analyses: The Dunn’s correction is not mentioned.

We thank the reviewer’s comment. We have added “followed by the Dunn’s multiple comparison test to compare each group” in Results section after description of the Kruskal-Wallis test.

  1. Results: Table 1. Please, change Gender with sex term. Gender is a psychological term not epidemiological/clinical.

According to the comment, we changed the word Gender to Sex.

  1. Gestational age (w) what “w” means?

According to the comment, we corrected the word to “weeks”.

  1. Table 3. Those no statistical differences between FeNO are hard to believe, could authors verify it?

We appreciate the Reviewer’s comment. This result was somewhat contrary to our expectations and we fully verified the results.

There have been conflicting results regarding FeNO in association with low and/or declining lung function. In observational studies of adult asthma, Matsunaga et al. reported that high FeNO was associated with progressive lung function decline (Matsunaga et al. Allergol Int 2016; 65: 266-271). Mogensen et al. reported that high urinary EDN, one of T2 inflammatory marker related to eosinophil activation, was associated with fixed airflow limitation, however, high FeNO was not. (Mogensen et al. Clin Exp Allergy 2019; 49: 155-162). In a recent systematic review, Ulrik et al. described FeNO as a promising biomarker to predict lung function decline in adult asthma. However, they also stated that the evidence is conflicting in children and young adults (Ulrik CS, et al. Eur Clin Respir J 2021; 8: 1891725).

As we stated as a limitation of our study in Discussion section, our retrospective cohort did not contain a group of patients whose symptoms had improved and who stopped visiting hospitals. Inclusion of the milder and outgrowing population would have yielded different results.

Overall, we believe that the current results should further be verified in a larger prospective cohort.

  1. References should be prepared according to journal recommendations.

We have revised the format of References accordingly.

Reviewer 2 Report

The authors presented a retrospective observational study of children treated for asthma in one of several specialized hospital in Japan, were followed for at least 5 years and had a least one PFT in 3 age periods namely 6-9. 10-12 and 13 to 15 y.o. Four lung function trajectories were defined with positive lung growth only in children with ‘upward trajectory’ as expected by their definition and a negative lung trajectory with was observed in all remaining groups including the ‘normal’ lung function group.

Major comments

·       No mention is made of obtaining ethics approval for this study. Permission to obtain data from the EMS and to merge the data from the 7 contributing hospitals must be obtained from the legal representative of the respective institutions and the ethics committee must have approved it.

·       To ascertain the possibility and magnitude of selection bias, it would be crucial to present a more detailed patient selection figure, starting with all children followed for asthma in one of the 7 hospitals with the n (%) of children who did not meet each eligibility criteria or have missing values.  In other words, what proportion of all children followed for asthma at these institutions, do the 253 children represent? Then present the group differences in baseline characteristics between those who were otherwise eligible but did not receive 3 spirometry testing (one in each age period) and those enrolled. The authors alluded to the possibility of such selection bias in the limitations but there is no data to help the reader assess the magnitude of such bias.

·       Authors should present the % of missing values for the determinants and how they dealt with them. It is impossible that they were no missing values.

·       Based on which reference values were % predicted values calculated? Did the reference values vary between hospital or over time? How was this controlled for?

·       It is not clear why lower limit of normal (LLN) was not used as cut-off rather than a cut-off value of 80%.

·       If not clear why FEV1/FVC ratio, a more sensitive marker of airway obstruction when FEV1 is in the normal rate, was not considered in the ascertainment of lung function trajectories. Probably a FEV1/FVC ratio below LLN with a FEV1 above LLN may have identify children with mild airway obstruction that were considered as having normal lung function.

·       How did the authors deal with patients who had more than 3 spirometry tests over the 9-year observations period? Presumably many patients have more than one spirometry test per 3-year period. Were all spirometry tests considered for the calculation of lung growth? If not, how were the spirometry test selected for this study among all those obtained in the 3-year age period? This would introduce a measurement bias.  

·       Was FeNO available in all patients at the same index visit as the spirometry? If not, what motivated the measurement of FeNo… and of spirometry?

·       Similarly, how were more than 3 FeNO or cACT /ACT measurements over the 9-year observations period dealt with? Were they all considered? If not, how were they selected? The authors should present the range of the number of measurements for lung function and of assessment of control (FeNO, cACT) per child. A child with more measurements per period is likely to have more opportunity for at least one abnormal measurement than a subject with only one such measurement. This would introduce a measurement bias.

·       Given the high rate of non-adherence, the data on prescribed medication should be taken with a grain of salt.

·       Is table 4 reporting a multivariate logistic regression adjusting for each co-variate or reported 5 independent bivariate associations? 

·       Discussion: one may not assume causality as a all measurements in a given age period appear to have been measured in cross-sectionally fashion. In other words, the treatment/determinant in 10-12 y could predict the outcome observed at 13-15 y.o. but not the treatment received (determinant measured) at the same index visit as spirometry at 13-15. in the latter case, it is an association that may well be confounding by indication.

Minor comments

·       Table 1, add p-value for the between-group comparisons since the study is not a randomized trial

·       Given the large range of biomarkers such as serum eosinophils, serum IgE, it may be more informative to classify children has having no (<150/uL), low (>150/uL) or high (>300/uL) eosinophilia, having an allergic status (total or specific IgE >=0.35) and test for any association with the outcome.

·       In table 2, if I understand the legend correctly, the p-value for anthropometric data refers to the comparison of the observed distribution with that expected of a normal distribution with a mean of 0 and SD of 1. Perhaps reporting children whose BMI fall in undernourished, normal, overweight, and obese children, would be useful to better display the patients’ characteristics. If I understand well all groups except those with upward lung function trajectory, had a mean BMI below 0 and there were no obese or overweight children in any groups? This is important for the generalizability of the study results to other populations.  It also raises the question as to the appropriateness of the reference values used for spirometry interpretation.

·       Did the trajectory of BMI over the 3 age periods differ between groups?

·       Refer to tables 1, 2, 3 and 4 at the appropriate places in the texts

·       Clarify, in Table 4, how Height Z-score in the 13-15 age period is binary.

Author Response

COMMENTS FROM REVIEWER #2:

The authors presented a retrospective observational study of children treated for asthma in one of several specialized hospitals in Japan, who were followed for at least 5 years and had a least one PFT in 3 age periods namely 6-9. 10-12 and 13 to 15 y.o. Four lung function trajectories were defined with positive lung growth only in children with ‘upward trajectory’ as expected by their definition and a negative lung trajectory with was observed in all remaining groups including the ‘normal’ lung function group.

We appreciate the Reviewer’s comments.

For clarification, the following words have been added to 1) Abstract and 2) Method section:

  • “---children with asthma -----at 7 specialized hospitals in Japan”
  • “This was a retrospective multicenter observational study”

Further, we apologize for the lack of explanation regarding the classification of lung function trajectories.

By definition, %FEV1 in normal range means actual measured values of FEV1 increased over the years (or grew) as the normal values increased with physical growth. Thus, FEV1 grew not only in Group U, but in Group N. In Group U, FEV1 further caught up in addition to the normal growth. On this rationale, we combined Group N and U.

We have modified the text in Methods section as follows:

“The subjects with %FEV1 ≥80 throughout the observation period were classified as Group N (normal trajectory), and those with %FEV1 <80 throughout the observation as Group L (low trajectory). Then, among the patients who had %FEV1<80 in any year of follow-up, those with S ≥0 were classified as Group U (upward trajectory) and those with S <0 as Group D (downward trajectory). In terms of lung function growth, FEV1 in Group N was considered to have grown or increased as normal values increased with physical growth. FEV1 in Group U was considered to have moved from relative decline to normal growth. Thus, favorable trajectories, i.e., Groups N and U, were then combined as Group N/U. FEV1 in Group L was considered to have not grown as the normal counterpart increased and FEV1 in Group D was considered to have further declined. Then, unfavorable trajectories, i.e., Groups L and D, were combined as Group L/D for analysis of risk”

Major comments

  1. No mention is made of obtaining ethics approval for this study. Permission to obtain data from the EMS and to merge the data from the 7 contributing hospitals must be obtained from the legal representative of the respective institutions and the ethics committee must have approved it.

We appreciate the Reviewer’s comment. Approval by the ethics committee has been described in Data collection section in Methods. However, as he/she pointed out, it is difficult to confirm the requisite in this place, so we created a new section, Ethics, and described as follows:

“The study was approved by the Ethics Committee of National Hospital Organization Mie National Hospital (approval number: 31-81) and permission to obtain data from the electronic medical systems and to merge the data from the 7 contributing hospitals, was granted.

  1. To ascertain the possibility and magnitude of selection bias, it would be crucial to present a more detailed patient selection figure, starting with all children followed for asthma in one of the 7 hospitals with the n (%) of children who did not meet each eligibility criteria or have missing values. In other words, what proportion of all children followed for asthma at these institutions, do the 253 children represent? Then present the group differences in baseline characteristics between those who were otherwise eligible but did not receive 3 spirometry testing (one in each age period) and those enrolled. The authors alluded to the possibility of such selection bias in the limitations but there is no data to help the reader assess the magnitude of such bias.

We thank the reviewer’s comment. First, we screened patients who had been followed up for 5 years or more and 629 patients were identified. According to our inclusion criteria, patients in whom spirometry was performed at least once during each of 3 age intervals: pre-adolescence (6 to 9 years of age), early adolescence (10 to 12 years of age) and adolescence (13 to 15 years of age). Now, we have revised Figure 1, the patient selection diagram. We examined the clinical background of patients who were included and excluded (new Table E1) and found that there were no differences in the background characteristics. As the reviewer pointed out, we did have more asthma patients at the institutions joined in the study. In our role as tertiary/secondary care institutions, however, we refer most patients from primary care institutions back to the referring institutions after a certain period of follow-up to establish a treatment plan. Long-term follow-up data for these patients were difficult to obtain and were therefore not included in this study. We have added the following sentence in the limitation paragraph in Discussion:

“In addition, our study population may not represent a whole pediatric asthma population since our institutions provide secondary/tertiary care and usually we refer patients back to referring primary institutions after a short period of evaluation and establishment of a treatment.”

  1. Authors should present the % of missing values for the determinants and how they dealt with them. It is impossible that they were no missing values.

We appreciate the reviewer’s comment. As for the background factors, the second column of Table 1 shows the number of data for each factor, from which the missing data can be read. Most were less than 10%. There was no missing data used for logistic analysis, which allowed us to include all the subjects in the analysis.

  1. Based on which reference values were % predicted values calculated? Did the reference values vary between hospitals or over time? How was this controlled for?

We utilized the reference equations for spirometry in Japanese children (6-18 years old) by the Japanese Society of Pediatric Pulmonology (reference 19). All spirometers available in Japan contain the equations. We have added the following sentence in Methods section as follows:

“, in which reference values were calculated by the equations for Japanese children.(19)”

  1. It is not clear why lower limit of normal (LLN) was not used as cut-off rather than a cut-off value of 80%.

We apologize for the lack of explanation. LLNs have been defined for Japanese children (reference 19). LLN for FEV1 is 80% and we employed the value. We revised it as follows:

“The subjects with %FEV1 ≥80, or above lower limit of normal (LLN) (19), throughout the observation period were classified as Group N (normal trajectory), and those with %FEV1 <80, or below LLN, throughout the observation as Group L (low trajectory).”

  1. If not clear why FEV1/FVC ratio, a more sensitive marker of airway obstruction when FEV1 is in the normal rate, was not considered in the ascertainment of lung function trajectories. Probably a FEV1/FVC ratio below LLN with a FEV1 above LLN may have identify children with mild airway obstruction that were considered as having normal lung function.

We appreciate the reviewer’s comment. As he/she pointed out, in cross-sectional studies on asthma severity in children, FEV1 % predicted values in most asthmatic children were reported to be within normal range and did not correlate well with symptom-defined severity of asthma, but FEV1/FVC ratio better correlated with severity (doi:10.1164/rccm.200308-1178OC, doi:10.1001/archpedi.162.12.1169). However, in terms of longitudinal changes in lung function, or lung function trajectories, most of the studies employ both FEV1 and FEV1/FVC ratio (there exist studies that employed only FEV1) and predictors identified with either parameter were similar. We then employed FEV1% predicted as a marker of airway narrowing.

  1. How did the authors deal with patients who had more than 3 spirometry tests over the 9-year observations period? Presumably many patients have more than one spirometry test per 3-year period. Were all spirometry tests considered for the calculation of lung growth? If not, how were the spirometry test selected for this study among all those obtained in the 3-year age period? This would introduce a measurement bias.

As the reviewer pointed out, there were patients who had spirometry measured more than 3 over the observation periods. We used all measurements for analysis as yearly average values, as stated in Methods section, “Spirometry results were evaluated as a 1-year average of percent predicted forced expiratory volume in one second (%FEV1)”.

  1. Was FeNO available in all patients at the same index visit as the spirometry? If not, what motivated the measurement of FeNo… and of spirometry?

FeNO was always measured when spirometry was performed.

  1. Similarly, how were more than 3 FeNO or cACT /ACT measurements over the 9-year observations period dealt with? Were they all considered? If not, how were they selected? The authors should present the range of the number of measurements for lung function and of assessment of control (FeNO, cACT) per child. A child with more measurements per period is likely to have more opportunity for at least one abnormal measurement than a subject with only one such measurement. This would introduce a measurement bias.

We employed the highest values of FeNO measured during each period. We apologize that we erroneously wrote as “mean” values and have corrected it accordingly.

ACT/c-ACT scores were also evaluated at every visit when spirometry was performed. We evaluated whether ACT/c-ACT fell into ≤19, “uncontrolled” levels, at least once during each period.(Line 182-183: “The number of patients who experienced loss of asthma control at any time during each period, represented as C-ACT or ACT ≤19”)

As the reviewer pointed out, variations in the number of measurements can cause bias. However, in our study, variations in the number of measurements were attributed to the number of visits during the follow-up periods, not the lack of measurements at regular visits. The study institutions as secondary/tertiary centers examined the patients at varying intervals, for example, some patients were seen every month and some patients, who were routinely treated at primary institutions, were seen every few months. Nonetheless, spirometry, FeNO and ACT/c-ACT were measured/evaluated at every visit to the study institutions.

To address the reviewer’s concern, we have added the number of measurements in Table E2 and the sentence in Methods as follows:

“Since the study institutions were secondary/tertiary centers, some patients were regularly treated at primary institutions and periodically referred from the institutions, intervals of spirometry, i.e. intervals of patient visits, varied. Numbers of spirometry measurements were recorded and summarized in Table E2.”

  1. Given the high rate of non-adherence, the data on prescribed medication should be taken with a grain of salt.

In our study subjects, adherence was adequately maintained in majority of the subjects. Although we did not measure adherence in quantitative way since there were no standardized measure of adherence in regular clinical practice, attending pediatricians recorded subjective evaluation of non-adherence. We found that the description of non-adherence was very few (9 in 273), if any, they returned to be adherent.
To address the reviewer’s concern, we have add the following sentences in Methods and Results:

Methods (lines 108-9), “Non-adherence recorded by the attending pediatricians was also retrieved.”

Results (lines 194-5), “Of note, non-adherence rate was adequately small at 9 in 273.”

  1. Is table 4 reporting a multivariate logistic regression adjusting for each co-variate or reported 5 independent bivariate associations?

Thank you for pointing out this. The analysis was a multivariate logistic regression adjusting for variables that were shown to be significantly different by univariate analyses as well as those considered clinically important as described in Results section (lines 196-200)

  1. Discussion: one may not assume causality as all measurements in a given age period appear to have been measured in cross-sectionally fashion. In other words, the treatment/determinant in 10-12 y could predict the outcome observed at 13-15 y.o. but not the treatment received (determinant measured) at the same index visit as spirometry at 13-15. in the latter case, it is an association that may well be confounding by indication.

We appreciate the reviewer’s comment. In our logistic model, factors may be associated with unfavorable trajectories over the 3 periods. Although confounding factors were adjusted in the model, associations found may not indicate consequent outcomes after preceding determinants. We have modified our description of limitations in Discussion as follows:

“Third, and most importantly, the risk factors identified in this study were merely shown to be associated with longitudinal changes in lung function, and causal relationships in the sequence of time were not demonstrated.”

Minor comments

  1. Table 1, add p-value for the between-group comparisons since the study is not a randomized trial

We thank the reviewer’s comment. We added p-values in Table 1.

  1. Given the large range of biomarkers such as serum eosinophils, serum IgE, it may be more informative to classify children has having no (<150/uL), low (>150/uL) or high (>300/uL) eosinophilia, having an allergic status (total or specific IgE >=0.35) and test for any association with the outcome.

We thank the reviewer for his/her constructive comment. Regarding allergic status, majority (264/273, 96.7%) of the patients were sensitized to HDM (ImmunoCAP >0.34) and we were not able to subcategorize the patients. Regarding blood eosinophil count, we categorized the data into <150 vs >150, <300 vs >300, and <500 vs >500, unfortunately, distributions of each category were similar among trajectory groups.

  1. In table 2, if I understand the legend correctly, the p-value for anthropometric data refers to the comparison of the observed distribution with that expected of a normal distribution with a mean of 0 and SD of 1. Perhaps reporting children whose BMI fall in undernourished, normal, overweight, and obese children, would be useful to better display the patients’ characteristics. If I understand well all groups except those with upward lung function trajectory, had a mean BMI below 0 and there were no obese or overweight children in any groups? This is important for the generalizability of the study results to other populations. It also raises the question as to the appropriateness of the reference values used for spirometry interpretation.

Anthropometry data in the majority of the subjects were within the normal range (<±2SD). There were no undernourished or obese children. Observed differences in anthropometric data were significant but very small. Pathophysiological significance needs further investigation. We have added a comment in Discussion as follows:

“  although pathophysiological significance of slightly short height needs further investigation.”

  1. Did the trajectory of BMI over the 3 age periods differ between groups?

The trajectory of BMI in each group was similar and we suspect that the anthropometric data represent constitutive characteristics of the subjects.

  1. Refer to tables 1, 2, 3 and 4 at the appropriate places in the texts

We apologized that Table 3 was not referred in the text and we corrected it.

  1. Clarify, in Table 4, how Height Z-score in the 13-15 age period is binary.

Height Z-score is continuous variable. Other variables are binary. We added the explanation accordingly.